# Maximum Likelihood Constraint Inference for Inverse Reinforcement Learning

**Dexter R.R. Scobee & S. Shankar Sastry**
Department of Electrical Engineering and Computer Sciences
University of California, Berkeley
{dscobee, sastry}@eecs.berkeley.edu

## Abstract

While most approaches to the problem of Inverse Reinforcement Learning (IRL) focus on estimating a reward function that best explains an expert agent's policy or demonstrated behavior on a control task, it is often the case that such behavior is more succinctly represented by a simple reward combined with a set of hard constraints. In this setting, the agent is attempting to maximize cumulative rewards subject to these given constraints on their behavior. We reformulate the problem of IRL on Markov Decision Processes (MDPs) such that, given a nominal model of the environment and a nominal reward function, we seek to estimate state, action, and feature constraints in the environment that motivate an agent's behavior. Our approach is based on the Maximum Entropy IRL framework, which allows us to reason about the likelihood of an expert agent's demonstrations given our knowledge of an MDP. Using our method, we can infer which constraints can be added to the MDP to most increase the likelihood of observing these demonstrations. We present an algorithm which iteratively infers the Maximum Likelihood Constraint to best explain observed behavior, and we evaluate its efficacy using both simulated behavior and recorded data of humans navigating around an obstacle.

## 1 Introduction

Advances in mechanical design and artificial intelligence continue to expand the horizons of robotic applications. In these new domains, it can be difficult to design a specific robot behavior by hand. Even manually specifying a task for a reinforcement-learning-enabled agent is notoriously difficult (Ho et al., 2015; Amodei et al., 2016). Inverse Reinforcement Learning (IRL) techniques can help alleviate this burden by automatically identifying the objectives driving certain behavior. Since first being introduced as Inverse Optimal Control by Kalman (1964), much of the work on IRL has focused on learning environmental rewards to represent the task of interest (Ng & Russell, 2000; Abbeel & Ng, 2004; Ratliff et al., 2006; Ziebart et al., 2008). While these types of IRL algorithms have proven useful in a variety of situations (Abbeel et al., 2007; Vasquez et al., 2014; Ziebart, 2010; Scobee et al., 2018), their basis in assuming that reward functions fully represent task specifications makes them ill suited to problem domains with hard constraints or non-Markovian objectives.

Recent work has attempted to address these pitfalls by using demonstrations to learn a rich class of possible specifications that can represent a task (Vazquez-Chanlatte et al., 2018). Others have focused specifically on learning constraints, that is, behaviors that are expressly forbidden or infeasible (Pardowitz et al., 2005; Pérez-D'Arpino & Shah, 2017; Subramani et al., 2018; McPherson et al., 2018; Chou et al., 2018). Such constraints arise in safety-critical systems, where requirements such as an autonomous vehicle avoiding collisions with pedestrians are more naturally expressed as hard constraints than as soft reward penalties. It is towards the problem of inferring such constraints that we turn our attention.

In this work, we present a novel method for inferring constraints, drawing primarily from the Maximum Entropy approach to IRL described by Ziebart et al. (2008). We use this framework to reason about the likelihood of observing a set of demonstrations given a nominal task description, as well as about their likelihood if we imposed additional constraints on the task. This knowledge allows us to select a constraint, or set of constraints, which maximizes the demonstrations' likelihood and

best explains the differences between expected and demonstrated behavior. Our method improves on prior work by being able to simultaneously consider constraints on states, actions and features in a Markov Decision Process (MDP) to provide a principled ranking of all options according to their effect on demonstration likelihood.

## 2 RELATED WORK

### 2.1 INVERSE REINFORCEMENT LEARNING

A formulation of the IRL problem was first proposed by Kalman (1964) as the Inverse problem of Optimal Control (IOC). Given a dynamical system and a control law, the author sought to identify which function(s) the control law was designed to optimize. This problem was brought into the domain of MDPs and Reinforcement Learning (RL) by Ng & Russell (2000), who proposed IRL as the task of, given an MDP and a policy (or trajectories sampled according to that policy), find a reward function with respect to which that policy is optimal.

One of the chief difficulties in the problem of IRL is the fact that a policy can be optimal with respect to a potentially infinite set of reward functions. The most trivial example of this is the fact that all policies are optimal with respect to a null reward function that always returns zero. Much of the work in IRL has been devoted to developing approaches that address this ambiguity by imposing additional structure to make the problem well-posed (Abbeel & Ng, 2004; Ratliff et al., 2006). Ziebart et al. (2008) approach the problem by employing the principle of maximum entropy (Jaynes, 1957), which allows the authors to develop an IRL algorithm that produces a single stochastic policy that matches feature counts without adding any additional constraints to the produced behavior. This so called Maximum Entropy IRL (MaxEnt) provides a framework for reasoning about demonstrations from experts who are noisily optimal. The induced probability distribution over trajectories forms the basis for our efforts in identifying the most likely behavior-modifying constraints.

### 2.2 BEYOND REWARD FUNCTIONS

While Markovian rewards do often provide a succinct and expressive way to specify the objectives of a task, they cannot capture all possible task specifications. Vazquez-Chanlatte et al. (2018) highlight the utility of non-Markovian Boolean specifications which can describe complex objectives (e.g. do this before that) and compose in an intuitive way (e.g. avoid obstacles *and* reach the goal). The authors of that work draw inspiration from the MaxEnt framework to develop their technique for using demonstrations to calculate the posterior probability that an agent is attempting to satisfy a Boolean specification.

A subset of these types of specifications that is of particular interest to us is the specification of constraints, which are states, actions, or features of the environment that must be avoided. Chou et al. (2018) explore how to infer trajectory feature constraints given a nominal model of the environment (lacking the full set of constraints) and a set of demonstrated trajectories. The core of their approach is to sample from the set of trajectories which have better performance than the demonstrated trajectories. They then infer that the set of possible constraints is the subset of the feature space that contains the higher-reward sampled trajectories, but not the demonstrated trajectories. Intuitively, they reason that if the demonstrator could have passed through those features to earn a higher reward, but did not, then there must have been a previously unknown constraint preventing that behavior. However, while their approach does allow for a cost function to rank elements from the set of possible constraints, the authors do not offer a mechanism for determining what cost function will best order these constraints.

Our approach to constraint inference from demonstrations addresses this open question by providing a principled ranking of the likelihood of constraints. We adapt the MaxEnt framework to allow us to reason about how adding a constraint will affect the likelihood of demonstrated behaviors, and we can then select the constraints which maximize this likelihood. We consider feature-space constraints as in Chou et al. (2018), and we explicitly augment the feature space with state- and action-specific features to directly compare the impacts of state-, action-, and feature-based constraints on demonstration likelihood.

## 3 MAXIMUM LIKELIHOOD CONSTRAINT INFERENCE

### 3.1 PROBLEM FORMULATION

Following the formulation presented by Ziebart et al. (2008), we base our work in the setting of a (finite-state) Markov Decision Process (MDP). We define an MDP $\mathcal{M}$ as a tuple $(S, \{A_s\}, \{P_{s,a}\}, D_0, \phi, R)$ where $S$ is a finite set of discrete states; $\{A_s\}$ is a set of the sets of actions available to be taken for each state $s$, such that $A_s \subseteq A$, where $A$ is a finite set of discrete actions; $\{P_{s,a}\}$ is a set of state transition probability distributions such that $P_{s,a}(s') = P(s'|s,a)$ is the probability of transitioning to state $s'$ after taking action $a$ from state $s$; $D_0 : S \to [0,1]$ is an initial state distribution; $\phi : S \times A \to \mathbb{R}^k_+$ is a mapping to a $k$-dimensional space of non-negative features; and $R : S \times A \to \mathbb{R}$ is a reward function. A trajectory $\xi$ through this MDP is a sequence of states $s_t$ and actions $a_t$ such that $s_0 \sim D_0$ and state $s_{i+1} \sim P_{s_i, a_i}$. Actions are chosen by an agent navigating the MDP according to a, potentially time-varying, policy $\pi$ such that $\pi(\cdot|s, t)$ is a probability distribution over actions in $A_s$. We denote a finite-time trajectory of length $T + 1$ by $\xi = \{\mathbf{s}_{0:T}, \mathbf{a}_{0:T}\}$.

At every time step $t$, a trajectory will accumulate features equal to $\phi(s_t, a_t)$. We use the notation $\phi_i(\cdot, \cdot)$ to refer to the $i$-th element of the feature map, and we use the label $\phi_i$ to denote the $i$-th feature itself. We also introduce an augmented indicator feature mapping $\widetilde{\phi}^{\mathbb{1}} : S \times A \to \{0,1\}^{n_\phi}$, where $n_\phi = k + |S| + |A|$. This augmented feature map uses binary variables to indicate the presence of a feature and expands the feature space by adding binary features to track occurrences of each state and action, such that

$$\widetilde{\phi}^{\mathbb{1}}_{\phi_i}(s,a) = \begin{cases} 1 & \text{if } \phi_i(s,a) > 0 \\ 0 & \text{otherwise} \end{cases}, \quad \widetilde{\phi}^{\mathbb{1}}_{s_i}(s,a) = \begin{cases} 1 & \text{if } s = s_i \\ 0 & \text{otherwise} \end{cases}, \quad \widetilde{\phi}^{\mathbb{1}}_{a_i}(s,a) = \begin{cases} 1 & \text{if } a = a_i \\ 0 & \text{otherwise} \end{cases}.$$

(1)

Typically, agents are modeled as trying to maximize, or approximately maximize, the total reward earned for a trajectory $\xi$, given by $R(\xi) = \sum_{t=0}^{T} \gamma^t R(s_t, a_t)$, where $\gamma \in (0, 1]$ is a discount factor. Therefore, an agent's policy $\pi$ is closely tied to the form of the MDP's reward function.

Conventional IRL focuses on inferring a reward function that explains an agent's policy, revealed through the behavior observed in a set of demonstrated trajectories $\mathcal{D}$. However, our method for constraint inference poses a different challenge: given an MDP $\mathcal{M}$, including a reward function, and a set of demonstrations $\mathcal{D}$, find the most likely set of constraints $C^*$ that could modify $\mathcal{M}$ to explain these demonstrations. We define our notion of constraints in the following section.

### 3.2 CONSTRAINTS FOR MDPS

Constraints are those behaviors that are not disallowed explicitly by the structure of the MDP, but which would be infeasible or prohibited for the underlying system being modeled by the MDP. This sort of discrepancy can occur when a generic or simplified MDP is designed without exact knowledge of specific constraints for the modeled system. For instance, for a generic MDP modeling the behavior of cars, we might want to include states for speeds up to 500km/h and actions for accelerations up to 12m/s$^2$. However, for a specific car on a specific roadway, the set of states where the vehicle travels above 100km/h may be prohibited because of a speed limit, and the set of actions where the vehicle accelerates above 4m/s$^2$ may be infeasible because of the physical limitations of the vehicle's engine. Therefore, any MDP trajectory of this specific car system would not contain a state-action pair which violates these legal and physical limits. Figure 1 shows an example of constraints driving behavior.

We define a constraint set $C_i \subseteq S \times A$ as a set of state-action pairs that violate some specification of the modeled system. We consider three general classes of constraints: state constraints, action constraints, and feature constraints. A state constraint set $C_{s_i} = \{(s,a) \mid s = s_i\}$ includes all state-action pairs such that the state component is $s_i$. An action constraint set $C_{a_i} = \{(s,a) \mid a = a_i\}$ includes all state-action pairs such that the action component is $a_i$. A feature constraint set $C_{\phi_i} = \{(s,a) \mid \phi_i(s,a) > 0\}$ includes all state-action pairs that produce a non-zero value for feature $\phi_i$.

If we augment the set of features as described in (1), it is straightforward to see that state and action constraints become special cases of feature constraints, with constraint sets given by

$C_i = \{(s, a) \mid \widetilde{\phi}_i^1(s, a) = 1\}$. It is also evident that we can obtain compound constraints, respecting two or more conditions, by taking the union of constraint sets $C_i$ to obtain $C = \bigcup_i C_i$.

### 3.2.1 ADDING CONSTRAINTS TO AN MDP

We need to be able to reason about how adding a constraint to an MDP will influence the behavior of agents navigating that environment. If we impose a constraint on an MDP, then none of the state-action pairs in that constraint set may appear in a trajectory of the constrained MDP. To enforce this condition, we must restrict the actions available in each state so that it is not possible for an agent to produce one of the constrained state-action pairs. For a given constraint $C$, we can replace the set of available actions $A_s$ in every state $s$ with an alternative set $A_s^C$ given by

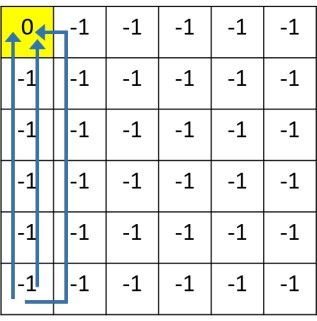

(a) Nominal MDP

$$A_s^C = A_s \setminus \{a \in A_s \mid (s, a) \in C\}. \qquad (2)$$

Performing such substitutions for an MDP $\mathcal{M}$ will lead to a modified MDP $\mathcal{M}^C$ such that $\mathcal{M}^C = (S, \{A_s^C\}, \{P_{s,a}\}, D_0, \phi, R)$.

The question then arises as to the how we should treat states with empty action sets $A_s^C = \emptyset$. Since an agent arriving in such an *empty state* would have no valid action to select, any trajectory visiting an empty state must be deemed invalid. Indeed, such empty action sets will be produced for any state $s_i$ such that $C_{s_i} \subseteq C$.

For MDPs with deterministic transitions, agents know precisely which state they will arrive in following a certain action. Therefore, any agent respecting constraints will not take an action that leads to an empty state, since doing so will lead to constraint violations. If we consider the set of empty states $S_{\text{empty}}$, then for the purposes of reasoning about an agent's behavior, we can impose an additional constraint set $C_{\text{empty}} = \{(s, a) \mid \exists s_{\text{empty}} \in S_{\text{empty}} : P_{s,a}(s_{\text{empty}}) = 1\}$. In this work, we will always implicitly add this constraint set, such that $\mathcal{M}^C$ will be equivalent to $\mathcal{M}^{C \cup C_{\text{empty}}}$, and we recursively add these constraints until reaching a fixed point.

For MDPs with stochastic transitions, the semantics of an empty state are less obvious and could lend themselves to multiple interpretations depending on the nature of the system being modeled. We offer a possible treatment in the appendix.

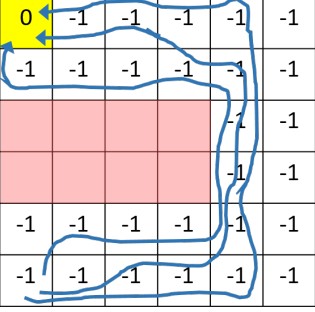

(b) Constrained MDP

Figure 1: Illustration of trajectories likely to be produced by noisily optimal agents navigating an MDP. (a) Expected behavior on a generic, nominal MDP. (b) Demonstrated behavior from a specific, constrained MDP. Numbers represent state-based rewards, and the red-shaded tiles represent state constraints.

### 3.2.2 NOMINAL MDPS

The nominal MDPs to which we will add constraints fall into two broad categories, which we denote as *generic* and *baseline*. The car MDP described at the beginning of Section 3.2 is an example of a generic nominal model: its state and action spaces are broad enough to encompass a wide range of car models, and we can use this nominal model to infer constraints that specialize the MDP to a specific car and task. For a generic nominal MDP, the reward function may also come from a generic, simplified task, such as "minimize time to the goal" or "minimize energy usage."

A baseline nominal MDP is a snapshot of a system from a point in time where it was well characterized. With a baseline nominal MDP, the constraints that we infer will represent changes to the system with respect to this baseline. In this case, the nominal reward function can be learned using existing IRL techniques with demonstrated behavior from the baseline model. We take this approach in our human obstacle avoidance example in Section 4.2: we use demonstrations of humans walking through the empty space to learn a nominal reward, then we can detect the presence of a new obstacle in the space from subsequent demonstrations.

### 3.3 Demonstration Likelihood Maximization

Our goal is to find the constraints $C^*$ which are most likely to have been added to a nominal MDP $\mathcal{M}$, given a set of demonstrations $\mathcal{D}$ from an agent navigating the constrained MDP. Let us define $P_{\mathcal{M}}$ to denote probabilities given that we are considering MDP $\mathcal{M}$. Our problem then becomes to select the constraints that maximize $P_{\mathcal{M}}(C \mid \mathcal{D})$. If we assume a uniform prior over possible constraints, then we know from Bayes' Rule that $P_{\mathcal{M}}(C \mid \mathcal{D}) \propto P_{\mathcal{M}}(\mathcal{D} \mid C)$. Therefore, in order to find the constraints that maximize $P_{\mathcal{M}}(C \mid \mathcal{D})$, we can solve the equivalent problem of finding which constraints maximize the likelihood of the given demonstrations. In this section, we present our approach to solving maximum likelihood constraint inference via solving demonstration likelihood maximization.

Under the maximum entropy model presented by Ziebart et al. (2008), the probability of a certain finite-length trajectory $\xi$ being executed by an agent traversing a deterministic MDP $\mathcal{M}$ is exponentially proportional to the reward earned by that trajectory.

$$P_{\mathcal{M}}(\xi) = \frac{1}{Z} e^{\beta R(\xi)} \mathbb{1}^{\mathcal{M}}(\xi), \tag{3}$$

where $Z$ is the *partition function*, $\mathbb{1}^{\mathcal{M}}(\xi)$ indicates if the trajectory is feasible for this MDP, and $\beta \in [0, \infty)$ is a parameter describing how closely an agent adheres to the task of optimizing the reward function (as $\beta \to \infty$, the agent becomes a perfect optimizer, and as $\beta \to 0$, the agent's actions become perfectly random). In the sequel, we assume that a given reward function will appropriately capture the role of $\beta$, so we omit $\beta$ from our notation without loss of generality.

In the case of finite horizon planning, the partition function will be the sum of the exponentially weighted rewards for all feasible trajectories on MDP $\mathcal{M}$ of length no greater than the planning horizon. We denote this set of trajectories by $\Xi_{\mathcal{M}}$. Because adding constraints $C$ modifies the set of feasible trajectories, we express this dependence as

$$Z(C) = \sum_{\xi \in \Xi_{\mathcal{M}}} e^{R(\xi)} \mathbb{1}^{\mathcal{M}^C}(\xi). \tag{4}$$

Assuming independence among demonstrated trajectories, the probability of observing a set $\mathcal{D}$ of $N$ demonstrations is given by the product

$$P_{\mathcal{M}^C}(\mathcal{D}) = \frac{1}{Z(C)^N} \prod_{\xi \in \mathcal{D}} e^{R(\xi)} \mathbb{1}^{\mathcal{M}^C}(\xi). \tag{5}$$

Our goal is to maximize the demonstration probability given by (5). Because we take the reward function and demonstrations as given, our only available decision variable in this maximization is the constraint set $C$ which alters the indicator $\mathbb{1}^{\mathcal{M}^C}$ and partition function $Z(C)$.

$$C^* = \underset{C \in \mathcal{C}}{\arg\max}\, P_{\mathcal{M}^C}(\mathcal{D}), \tag{6}$$

where $\mathcal{C} \subseteq 2^{S \times A}$ is the hypothesis space of possible constraints.

From the form of (5), it is clear that to solve (6), we must choose a constraint set that does not invalidate any demonstrated trajectory while simultaneously minimizing the value of $Z(C)$. Consider the set of trajectories that would be made *infeasible* by augmenting the MDP with constraint $C$, which we denote as $\Xi^-_{\mathcal{M}^C} = \{\xi \in \Xi_{\mathcal{M}} \mid \mathbb{1}^{\mathcal{M}^C}(\xi) = 0\}$. The value of $Z(C)$ is minimized when we maximize the sum of exponentiated rewards of these infeasible trajectories. Considering the form of the trajectory probability given by (3), we can see that this sum is proportional to the total probability of observing a trajectory from $\Xi^-_{\mathcal{M}^C}$ on the original MDP $\mathcal{M}$

$$\sum_{\xi \in \Xi^-_{\mathcal{M}^C}} e^{R(\xi)} \propto \sum_{\xi \in \Xi^-_{\mathcal{M}^C}} P_{\mathcal{M}}(\xi) = P_{\mathcal{M}}(\Xi^-_{\mathcal{M}^C}). \tag{7}$$

This insight leads us to the final form of the optimization

$$\begin{aligned} C^* = \underset{C \in \mathcal{C}}{\arg\max}\, & P_{\mathcal{M}}(\Xi^-_{\mathcal{M}^C}) \\ \text{s.t. } & \mathcal{D} \cap \Xi^-_{\mathcal{M}^C} = \emptyset \end{aligned}. \tag{8}$$

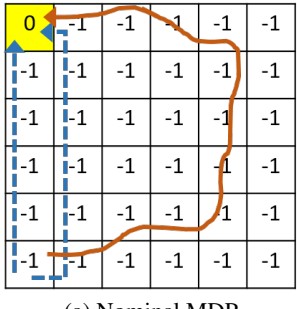 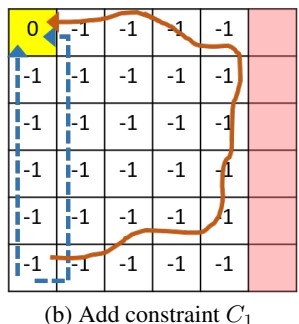 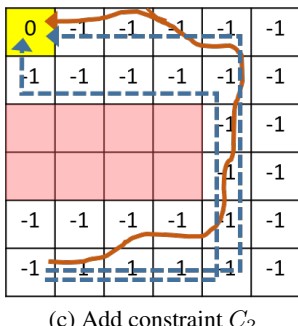

(a) Nominal MDP      (b) Add constraint $C_1$      (c) Add constraint $C_2$

Figure 2: Selecting constraints to maximize demonstration likelihood. Trajectories that are likely to be observed on a given MDP are shown as dashed, angular arrows, and a provided demonstration is shown as a solid, curved arrow. Adding $C_1$ in (b) does little to align the expected trajectories with the demonstration. On the other hand, adding $C_2$ in (c) makes the original expected trajectories infeasible and causes the new expected trajectories to agree with the demonstration, greatly increasing the likelihood of the demonstration on this constrained MDP.

In order to solve (8), we must reason about the probability distribution of trajectories on the original MDP $\mathcal{M}$, then find the constraint $C$ such that $\Xi^-_{\mathcal{M}^C}$ contains the most probability mass while not containing any demonstrated trajectories. Figure 2 provides a graphical representation of this reasoning. We highlight here that the fact that the chosen $C$ must not conflict with *any* demonstration is an important condition: *all* provided demonstrations must perfectly respect a constraint in order for it to be learned, otherwise a less restrictive set of constraints may be learned instead. Future work will look to relax this requirement in order to learn about constraints that are generally respected by a set of demonstrations, without needing to first isolate just the successful, constraint-respecting demonstrations.

While equation (8) is derived for deterministic MDPs, if we can assume, as proposed by Ziebart et al. (2008), that for a given stochastic MDP, the stochastic outcomes have little effect on an agent's behavior and the partition function, then the solution to (8) will also approximate the optimal constraint selection for that MDP. However, in order to fully address the stochastic case, we would need to reformulate our approach based on maximum *causal* entropy (Ziebart, 2010). We save this extension for future work.

### 3.3.1 CONSTRAINT HYPOTHESIS SPACE

In order for the solutions to (8) to be meaningful, we must be careful with our choice of the constraint hypothesis space $\mathcal{C}$. For instance, if we let $\mathcal{C} = 2^{S \times A}$, then the optimal solution will always be to choose the most restrictive $C$ to constrain *all* state-action pairs not observed in the demonstration set.

One approach to avoid this trivial solution is to use domain knowledge of the modeled system to restrict $\mathcal{C}$ to a library of plausible or common constraints. McPherson et al. (2018) construct such a library by using reachability theory to calculate a family of likely unsafe sets.

We could also potentially address this problem by adding a regularization term to the optimization that would penalize constraints based on some notion of "size," which would encourage the selection of "smaller" constraints. The size of a constraint set could be defined by the number of *minimal constraints* that it contains. These minimal constraint sets constrain a single state, action, or feature, and were introduced in Section 3.2 as $C_{s_i}$, $C_{a_i}$, and $C_{\phi_i}$, respectively. While penalizing this definition of size would effectively discourage overfitting, considering every possible combination of minimal constraints causes the hypothesis space to grow exponentially in the number of states, actions, and features of the MDP, which may make directly solving this formulation intractable.

Another approach, which avoids this combinatorial explosion, is to use the minimal constraint sets themselves, not their combinations, as our hypothesis space, and to select from among these constraints in an iterative, greedy manner. By iteratively selecting individual minimal constraint sets, and choosing a proper stopping condition, it is possible to gradually grow the full estimated constraint

set and avoid overfitting to the demonstrations. It is this method that we will utilize in this work. Section 3.4 details our approach for selecting the most likely minimal constraint, and Section 3.5 details our approach for iteratively growing the estimated constraint set.

## 3.4 Probability Mass for Minimal Constraints

As detailed in Section 3.3, the most likely constraint set is the one whose eliminated trajectories $\Xi^-_{\mathcal{M}^C}$ have the highest probability of being demonstrated on the original, unconstrained MDP. Therefore, to find the most likely of the minimal constraints, we must find the expected proportion of trajectories which will contain any state or action, or accrue any feature. By using our augmented indicator feature map from (1), we can reduce this problem to only examine feature accruals. Ziebart et al. (2008) present their forward-backward algorithm for calculating expected feature counts for an agent following a policy in the maximum entropy setting. This algorithm nearly suffices for our purposes, but it computes the expectation of the total number of times a feature will be accrued (i.e. how often will this feature be observed per trajectory), rather than the expectation of the number of trajectories that will accrue that feature at *any* point in time. To address this problem, we present a modified form of the "forward" pass as Algorithm 1. Our algorithm tracks state visitations as well as feature accruals at each state, which allows us to produce the same maximum entropy distribution over trajectories as Ziebart et al. (2008) while not counting additional accruals for trajectories that have already accrued a feature.

---

**Algorithm 1** Feature Accrual History Calculation

**Input:** an MDP $\mathcal{M}$, a policy $\pi(a|s,t)$, a time horizon $T$
**Output:** expected feature accrual history $\widetilde{\Phi}_{[1,T]}$
/* Initialize state visitation and feature accrual history */
1: **for** $s \in S$ **do**
2:     $D_{s,0} \leftarrow D_0(s)$
3:     $\widetilde{\Phi}_{s,0} \leftarrow \mathbf{0}_{n_\phi \times 1}$
4: **end for**
/* Track feature accruals over the time horizon */
5: **for** $t \in [0,\ T-1]$ **do**
6:     **for** $s \in S$ **do**
7:       **for** $a \in A_s$ **do**
8:         /* New feature accruals */
9:         /* "$\odot$" denotes element-wise multiplication */
10:         $\Delta\widetilde{\Phi}_{s,t}(a) \leftarrow \widetilde{\phi}^{\mathbb{1}}(s,a) \odot \left( D_{s,t}\mathbf{1}_{n_\phi \times 1} - \widetilde{\Phi}_{s,t} \right)$
11:       **end for**
12:     **end for**
13:     **for** $s' \in S$ **do**
14:       $D_{s',t+1} \leftarrow \sum\limits_{s \in S} \sum\limits_{a \in A_s} D_{s,t}\pi(a|s,t)P(s'|s,a)$
15:       $\widetilde{\Phi}_{s',t+1} \leftarrow$
          $\sum\limits_{s \in S} \sum\limits_{a \in A_s} \left( \widetilde{\Phi}_{s,t} + \Delta\widetilde{\Phi}_{s,t}(a) \right) \pi(a|s,t)P(s'|s,a)$
16:     **end for**
17:     $\widetilde{\Phi}_{t+1} \leftarrow \sum\limits_{s \in S} \widetilde{\Phi}_{s,t+1}$
18: **end for**
19: Return $\widetilde{\Phi}_{[1,T]}$

---

The input of Algorithm 1 includes the MDP itself, a time horizon, and a time-varying policy. This policy should capture the expected behavior of the demonstrator on the nominal MDP $\mathcal{M}$, and it can be computed via the "backward" part of the algorithm from Ziebart et al. (2008). The output of Algorithm 1, $\widetilde{\Phi}_{[1,T]}$, is an $n_\phi \times T$ array such that the $t$-th column $\widetilde{\Phi}_t$ is a vector whose $i$-th entry is the expected proportion of trajectories to have accrued the $i$-th feature by time $t$. In particular, the $i$-th element of $\widetilde{\Phi}_T$ is equal to $P_{\mathcal{M}}(\Xi^-_{\mathcal{M}^{C_i}})$, which allows us to now directly select the most likely constraint according to (8).

## 3.5 Maximum-Coverage-Based Iterative Constraint Inference

When using minimal constraint sets as the constraint hypothesis space, it is possible that the most likely constraint still does not provide a satisfactory explanation for the demonstrated behavior. In this case, it can be beneficial to combine minimal constraints. If the task of solving (8) is framed as finding the combination of constraint sets that "covers" the most probability mass, then the problem becomes a direct analog for the classic maximum coverage problem. While this problem is known to be NP-hard, there exist a simple greedy algorithm with known suboptimality bounds (Hochbaum & Pathria, 1998).

We present Algorithm 2 as our approach for adapting this greedy heuristic to solve the problem of constraint inference. At each iteration, we grow our estimated constraint set by augmenting it with the constraint set in our hypothesis space that covers the most *currently uncovered* probability mass. By analogy to the maximum coverage problem (Hochbaum & Pathria, 1998), we derive the following bound on the suboptimality of our approach.

**Theorem 1.** *Let $\mathcal{C}_{n_c}$ be the set of all constraints $\mathbf{C}_{n_c}$ such that $\mathbf{C}_{n_c} = \bigcup_{i=1}^{n_c} C_i$ for $C_i \in \mathcal{C}$, and let $\mathbf{C}_{n_c}^*$ be the solution to (8) using $\mathcal{C}_{n_c}$ as the constraint hypothesis space. It follows, then, that at the end of every iteration $i$ of Algorithm 2,*

$$P\left(\Xi^-_{\mathcal{M}^{\widehat{C}^*}}\right) \geq \left(1 - \left(\frac{i-1}{i}\right)^i\right) P\left(\Xi^-_{\mathcal{M}^{\mathbf{C}_i^*}}\right).$$

This bound is directly analogous to the suboptimality bound for the greedy solution to the maximum coverage problem proven by Hochbaum & Pathria (1998). For space, the proof is included in the appendix.

Rather than selecting the number of constraints $n_c$ to be used ahead of time, we check a stopping condition to decide if we should continue to add constraints. Because we are attempting to maximize $P_{\mathcal{M}^C}(\mathcal{D})$, it might seem natural to use a probability-based stopping condition. However, choosing a stopping criterion based on probability is problematic because

---

**Algorithm 2** Greedy Iterative Constraint Inference

**Input:** MDP $\mathcal{M}$, constraint hypothesis space $\mathcal{C}$, empirical probability distribution $P_{\mathcal{D}}$, threshold $d_{D_{\mathrm{KL}}}$

**Output:** estimated constraint set $\widehat{C}^*$

1: $\widehat{C}^* \leftarrow \emptyset$
2: **for** $i \in [1, |\mathcal{C}|]$ **do**
3: $\quad C_i \leftarrow$ solution to (8) using $\mathcal{M}^{\widehat{C}^*}$, $\mathcal{C}$, and $\mathcal{D}$
4: $\quad \Delta_{D_{\mathrm{KL}}} = D_{\mathrm{KL}}\left(P_{\mathcal{D}} \,\|\, P_{\mathcal{M}^{\widehat{C}^*}}\right) - D_{\mathrm{KL}}\left(P_{\mathcal{D}} \,\|\, P_{\mathcal{M}^{\widehat{C}^* \cup C_i}}\right)$
5: $\quad$ **if** $\Delta_{D_{\mathrm{KL}}} \leq d_{D_{\mathrm{KL}}}$ **then**
6: $\qquad$ **break**
7: $\quad$ **end if**
8: $\quad \widehat{C}^* \leftarrow \widehat{C}^* \cup C_i$
9: **end for**
10: Return $\widehat{C}^*$

---

the probability of observing a set of demonstrations is dependent on the number of demonstrations in the set, even if each individual demonstration has the same probability. We instead base our stopping condition on KL divergence, which depends only on the distribution of trajectories in $\mathcal{D}$ and not on the number of demonstrations. The quantity $D_{\mathrm{KL}}\left(P_{\mathcal{D}} \,\|\, P_{\mathcal{M}^{\widehat{C}^*}}\right)$ provides a measure of how well the distribution over trajectories induced by our inferred constraints, $P_{\mathcal{M}^{\widehat{C}^*}}$, agrees with the empirical probability distribution over trajectories observed in the demonstrations, $P_{\mathcal{D}}$. Using this KL divergence in the stopping condition actually preserves a link to the probability of the demonstration set, since the KL divergence will decrease monotonically as $P_{\mathcal{M}^C}(\mathcal{D})$ increases. The threshold parameter $d_{D_{\mathrm{KL}}}$ is chosen to avoid overfitting to the demonstrations, combating the tendency to select additional constraints that may only marginally better align our predictions with the demonstrations.

## 4 EXAMPLES

### 4.1 SYNTHETIC GRID WORLD

We consider the grid world MDP presented in Figure 3. The environment consists of a 9-by-9 grid of states, and the actions are to move up, down, left, right, or diagonally by one cell. The objective is to move from the starting state in the bottom-left corner ($s_0$) to the goal state in the bottom-right corner ($s_G$). Every state-action pair produces a distance feature, and the MDP reward is negative distance, which encourages short trajectories. There are additionally two more features, denoted green and blue, which are produced by taking actions from certain states, as shown in Figure 3.

The true MDP, from which agents generate trajectories, is shown in Figure 3a, including its constraints. The nominal, more generic MDP shown in Figure 3b is what we take as $\mathcal{M}$ for applying the iterative maximum likelihood constraint inference in Algorithm 2, with feature accruals estimated using Algorithm 1. While Figures 3c through 3e show the iteratively estimated constraints, which align with the true constraints, it is interesting to note that not all constraints present in the true MDP are identified. For instance, it is so unlikely that an agent would ever select the up-left diagonal action, that the fact that demonstrated trajectories did not contain that action is unsurprising and does not make that action an estimated constraint.

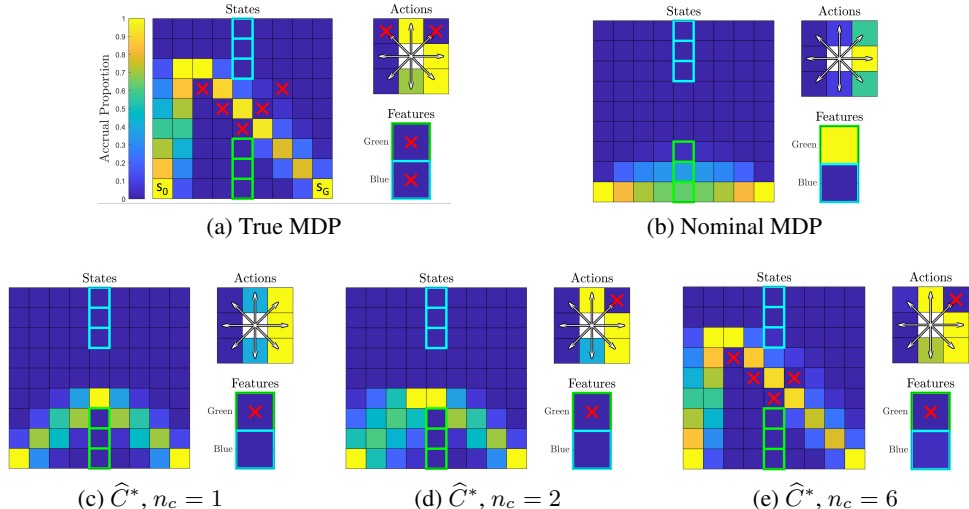

(a) True MDP

(b) Nominal MDP

(c) $\widehat{C}^*$, $n_c = 1$

(d) $\widehat{C}^*$, $n_c = 2$

(e) $\widehat{C}^*$, $n_c = 6$

Figure 3: Algorithm performance on a synthetic grid world MDP. Each subfigure represents the MDP by showing (clockwise from left) its states, actions, and features. Each element is shaded according to the proportion of trajectories that are expected to accrue the respective augmented feature, computed via Algorithm 1. Constraints are marked with a red "X," and bright bounding boxes mark the green and blue feature-producing states. The result here are shown for a set of 100 demonstrations sampled according to the expectation for the True MDP (a). We begin with the nominal MDP shown in (b), and produce (c), (d), and (e) by applying Algorithm 2. Note that (c), (d), and (e) show the selections of feature, action, and state constraints, respectively.

Figure 4 shows how the performance of our approach varies based on the number of available demonstrations and the selection for the threshold $d_{D_{\mathrm{KL}}}$. The false positive rate shown in Figure 4a is the proportion of selected constraints which are not constraints of the true system. We can observe two trends in this data that we would expect. First, lower values of $d_{D_{\mathrm{KL}}}$ lead to greater false positive rates since they allow Algorithm 2 to continue iterating and accept constraints that do less to align expectations and demonstrations. Second, having more demonstrations available provides more information and reduces the rate of false positives. Further, Figure 4b shows that more demonstrations also allows the behavior predicted by constraints to better align with the observations. It is interesting to note, however, that with fewer than 10 demonstrations and a very low $d_{D_{\mathrm{KL}}}$, we may produce very low KL divergence, but at the cost of a high false positive rate. This phenomenon highlights the role of selecting $d_{D_{\mathrm{KL}}}$ to avoid over-fitting. The threshold $d_{D_{\mathrm{KL}}} = 0.1$ achieves a good balance of producing few false positives with sufficient examples while also producing lower KL divergences, and we used this threshold to produce the results in Figures 3 and 5.

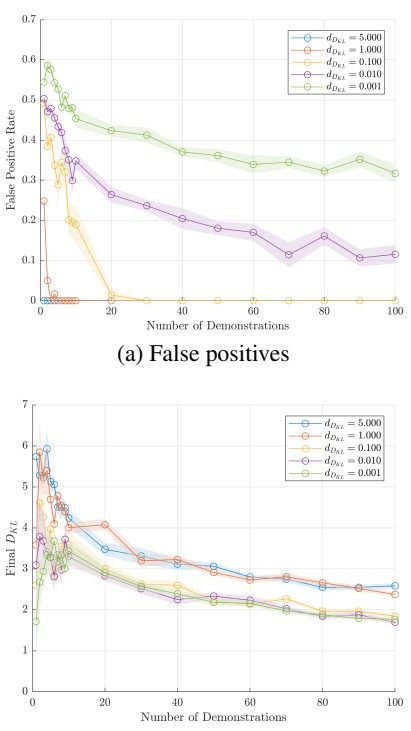

(a) False positives

(b) $D_{KL}$

Figure 4: Algorithm performance on the synthetic grid world. Each data point represents the mean result of 10 independent trajectory draws, and the margins show $\pm 1$ standard error.

## 4.2 HUMAN OBSTACLE AVOIDANCE

In our second example, we analyze trajectories from humans as they navigate around an obstacle on the floor. We map these continuous trajectories into trajectories through a grid world where each cell represents a 1ft-by-1ft area

on the ground. The human agents are attempting to reach a fixed goal state ($s_G$) from a given initial state ($s_0$), as shown in Figure 5. We performed MaxEnt IRL on human demonstrations of the task without the obstacle to obtain the nominal distance-based reward function. We restrict ourselves to estimating only state constraints, as we do not supply our algorithm with knowledge of any additional features in the environment and we assume that the humans' motion is unrestrained.

Demonstrations were collected from 16 volunteers, and the results of performing constraint inference are shown in Figure 5. Our method is able to successfully predict the existence of a central obstacle. While we do not estimate every constrained state, the constraints that we do estimate make all of the obstacle states unlikely to be visited. In order to identify those states as additional constraints, we would have to decrease our $d_{D_{\mathrm{KL}}}$ threshold, which could also lead to more spurious constraint selections, such as the three shown in Figure 5.

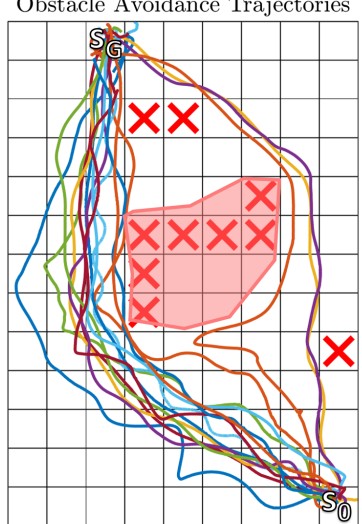

Figure 5: Human trajectories overlaid on a grid world MDP. The shaded region represents an obstacle in the human's environment, and the red "X"s represent learned constraints.

## 5 CONCLUSION AND FUTURE WORK

We have presented our novel technique for learning constraints from demonstrations. We improve upon previous work in constraint-learning IRL by providing a principled framework for identifying the *most likely* constraint(s), and we do so in a way that explicitly makes state, action, and feature constraints all directly comparable to one another. We believe that the numerical results presented in Section 4 are promising and highlight the usefulness of our approach.

Despite its benefits, one drawback of our approach is that the formulation is based on (3), which only exactly holds for deterministic MDPs. As mentioned in Section 3.3, we plan to investigate the use of a maximum *causal* entropy approach to address this issue and fully handle stochastic MDPs. Additionally, the methods presented here require *all* demonstrations to contain *no* violations of the constraints we will estimate. We believe that softening this requirement, which would allow reasoning about the likelihood of constraints that are occasionally violated in the demonstration set, may be beneficial in cases where trajectory data is collected without explicit labels of success or failure. Finally, the structure of Algorithm 1, which tracks the expected features accruals of trajectories over time, suggests that we may be able to reason about non-Markovian constraints by using this historical information to our advantage.

Overall, we believe that our formulation of maximum likelihood constraint inference for IRL shows promising results and presents attractive avenues for further investigation.

### ACKNOWLEDGMENTS

This work is supported by the National Science Foundation through grant CNS-1545126 (VeHICaL). We would like to thank Jaime Fisac, Andrea Bajcsy, Sylvia Herbert, and David Fridovich-Keil for their insightful discussions and for sharing their collected data.

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

APPENDIX

## A    ADDING CONSTRAINTS TO STOCHASTIC MDPs

For MDPs with stochastic transitions, the semantics of an empty state are less obvious and could lend themselves to multiple interpretations depending on the nature of the system being modeled. In our context, we use constraints to describe how observed behaviors from *demonstrations* differ from possible behaviors allowed by the nominal MDP structure. We therefore assume that any demonstrations provided are, by the fact that they were selected to be provided, consistent with the system's constraints, including avoiding empty states. This assumption implies that any stochastic state transitions that would have led to an empty state will not be observed in trajectories from the demonstration set. The omission of these transitions means that, for a given $(s, a)$, if $P_{s,a}(S_{\text{empty}}) = p$, then a proportion $p$ of these $(s, a)$ pairs which occur as an agent navigates the environment will be excluded from demonstrations. Therefore, as we modify the MDP to reason about demonstrated behavior, we need updated transition probabilities which eliminate the probability mass of transitioning to empty states, an event which will never be observed in a demonstration. Such modified probabilities can be given as

$$P_{s,a}^C(s') = \begin{cases} 0 & \text{if } s' \in S_{\text{empty}} \\ \frac{Ps,a(s')}{1 - Ps,a(S_{\text{empty}})} & \text{otherwise} \end{cases}. \tag{9}$$

We must also capture the change to observed state-action pair frequencies by understanding that any observed policy $\pi^C$ will be related to an agent's actual policy $\pi$ according to

$$\pi^C(a|s, t) = \frac{\pi(a|s, t)(1 - P_{s,a}(S_{\text{empty}}))}{\sum\limits_{a' \in A_s^C} \pi(a'|s, t)(1 - P_{s,a}(S_{\text{empty}}))}. \tag{10}$$

It is important to note that the modifications presented in (9) and (10) for non-deterministic MDPs are not meant to directly reflect the reality of the underlying system (we wouldn't expect the actual transition dynamics to change, for instance), but to reflect the *apparent* behavior that we would expect to observe in the subset of trajectories that would be selected as demonstrations. We further note that applying these modifications to deterministic MDPs will result in the same expected behavior as augmenting the constraint set with $C_{\text{empty}}$.

## B    PROOF FOR THEOREM 1

**Theorem 1.** *Let $\mathcal{C}_{n_c}$ be the set of all constraints $\mathbf{C}_{n_c}$ such that $\mathbf{C}_{n_c} = \bigcup_{i=1}^{n_c} C_i$ for $C_i \in \mathcal{C}$, and let $\mathbf{C}_{n_c}^*$ be the solution to (8) using $\mathcal{C}_{n_c}$ as the constraint hypothesis space. It follows, then, that at the end of every iteration $i$ of Algorithm 2,*

$$P\left(\Xi_{\mathcal{M}\widehat{C}^*}^-\right) \geq \left(1 - \left(\frac{i-1}{i}\right)^i\right) P\left(\Xi_{\mathcal{M}\mathbf{c}_i^*}^-\right).$$

*Proof.* The problem of finding $\mathbf{C}_{n_c}^*$ is analogous to solving the maximum coverage problem, discussed by Hochbaum & Pathria (1998), where the set of elements to be covered is the set of trajectories $\{\xi \mid \exists C \in \mathcal{C} : \xi \in \Xi_{\mathcal{M}C}^- \text{ and } \mathcal{D} \cap \Xi_{\mathcal{M}C}^- = \emptyset\}$ and the weight of each element $\xi$ is $P_{\mathcal{M}}(\xi)$. Because Algorithm 2 constructs $\widehat{C}^*$ iteratively by taking the union of the previous value of $\widehat{C}^*$ and the set $C_i \in \mathcal{C}$ which solves (8), the value of $\widehat{C}^*$ at the end of the $i$-th iteration is analogous to the greedy solution of the maximum coverage problem with $n_c = i$. Therefore, we can directly apply the suboptimality bound for the greedy solution proven in Hochbaum & Pathria (1998) to arrive at our given bound on eliminated probability mass. ∎

