# OpenReview forum: "Maximum Likelihood Constraint Inference for Inverse Reinforcement Learning"
_ICLR.cc/2020/Conference — Accept (Spotlight)_

### Official Review · AnonReviewer3 · 2019-10-24
**Official Blind Review #3**

**Rating:** 6

**Review:**

The submission considers estimating the constraints on the state, action and feature in the provided demonstrations, instead of learning rewards. The authors use the likelihood as MaxEnt IRL methods to evaluate the "correctness" of the constraints, and find the most likely constraints given the demonstrations. While the problem is challenging (NP-hard), suboptimality of the proposed algorithm is analyzed. Experiments are provided to demonstrate the performance of the proposed method.

The problem considered is interesting, and the authors provide a straightforward but empirically effective method. However, the motivation is a little unclear to me. Specifically, what will be the practical cases, where the learning the constraints is important and necessary? Can authors further motivate this topic by providing more real-world applications?



**Experience Assessment:**

I do not know much about this area.

**Review Assessment: Checking Correctness Of Derivations And Theory:**

I assessed the sensibility of the derivations and theory.

**Review Assessment: Checking Correctness Of Experiments:**

I assessed the sensibility of the experiments.

**Review Assessment: Thoroughness In Paper Reading:**

I read the paper at least twice and used my best judgement in assessing the paper.

---

> ### Author Response · Authors · 2019-11-14
> **Response to Review #3**
>
> Thank you for your review and for pointing out that this is an interesting problem to address! We address your question about applications below, and we’ve added a bit of these ideas to the introduction of the updated version as well.
>
> To continue with the car example introduced in the text, it could be possible to automatically learn about changes in road accessibility, such as temporary obstacles or detours, when other cars unexpectedly avoid an area and take seemingly sub-optimal routes. Moreover, in driving tasks, collisions are an unacceptable behavior, so we argue that they are better modeled by hard constraints than by soft penalties in the reward function. More broadly speaking, constraints are an important part of defining safe behavior in safety-critical systems. For instance, we would certainly want a robot learning to assist with surgical tasks to be able to learn that certain actions must always be avoided.
>
> Another specific application that we’re interested in exploring is in automated diagnosis of human motor impairments. The nominal model can be based on the range of motion / abilities of healthy individuals (or a baseline from a patient), and demonstrations can be provided by patients over time. Our method could then detect a decline in motor abilities (i.e. the presence of new constraints) and assist clinicians in inferring if an individual’s motor impairment is more likely to be caused by limited joint ranges (state constraint) or loss of strength (action constraint).

---

### Official Review · AnonReviewer1 · 2019-10-24
**Official Blind Review #1**

**Rating:** 3

**Review:**

The paper considers learning of constraints in MDPs in an IRL setting with the goal of maximizing the likelihood of demonstrations (in the constrained MDP). The constraints come in the form of avoiding certain states, actions or features. The authors propose an algorithm for learning the constraints and evaluate their approach in synthetic and a real-world experiment.

The paper is mainly well written and considers a relevant problem. However, at a conceptual level, I am missing a more precise problem formulation and an extended discussion of the possible failure cases/limitations of the approach. Also more real-world experiments would be welcome (the presented real-world problem is relatively easy, expert's trajectories are likely to agree and not make any "mistakes").

In more detail, I think the paper can be improved by spelling out the problem formulation more precisely (Section 3.1) -- in its current form I think it somewhat fuzzy. What I mean by this is that there is a nominal MDP (which is at least as "large" as the true MDP) and there are demonstrations. The goal seems to be to identify constraints from the demonstrations and the MDP such that if added to the nominal MDP it "shrinks" to the true MDP. Is that the actual underlying problem? (I understand the current formulation in the terms of the likelihood of demonstrations). In this formulation, I think the definition of over-fitting arises naturally. In the current formulation though, over-fitting seems somewhat disconnected in the sense that even if we identify a solution which generalizes perfectly to new demonstrations, we would talk about over-fitting (the optimization of the likelihood maximization is not the problem that we actually want to solve).

Further, as also mentioned by the authors, there is a problem with sub-optimal demonstrators. Considering the example of cars given in the paper, there might be drivers that do violate speed limits. In that case, the proposed approach will fail to identify some constraints. On the other hand, if all "optimal" demonstrations are to go fast, the approach would constraint the possibility to drive slowly. All this is fine, but I think it warrants a broader discussion which is not deferred to the Conclusion/Future Work section as it is very crucial regarding the applicability of the proposed approach.



**Experience Assessment:**

I have published one or two papers in this area.

**Review Assessment: Checking Correctness Of Derivations And Theory:**

I assessed the sensibility of the derivations and theory.

**Review Assessment: Checking Correctness Of Experiments:**

I assessed the sensibility of the experiments.

**Review Assessment: Thoroughness In Paper Reading:**

I read the paper at least twice and used my best judgement in assessing the paper.

---

> ### Author Response · Authors · 2019-11-14
> **Response to Review #1**
>
> Thank you for your detailed and constructive feedback! Discussing these points will help us clarify and improve the paper. You seem to raise two main points: looking for more detail in the problem formulation and looking for expanded discussion on the limitations of the approach.
>
> To your point on the problem formulation: you are correct that our goal is to try to identify constraints from demonstrations and the nominal MDP to attempt to recover the true MDP, and we believe that it is a fair and useful analogy to think of adding these constraints as “shrinking” the nominal MDP to the true MDP. We approach this problem from the perspective of selecting the most likely constraints (to shrink the MDP) given the available information, namely the nominal MDP and the demonstrations. If we take the nominal model as given and assume a uniform prior over possible constraints, then choosing the most likely constraints given the demonstrations is equivalent to choosing the constraints that maximize the likelihood of the demonstrations, which is why we pursue this formulation. We’ve added some additional language at the end of sec. 3.1 and beginning of sec. 3.3 to clarify that maximizing demonstrations likelihood is not the goal itself, but an approach to achieve the goal of finding the most likely constraints by solving an equivalent problem.
>
> With regards to your comment on over-fitting, we believe that this term is still well suited to the demonstration likelihood maximization formulation we present. For example, if we learn constraints from a training set of demonstrations, but these constraints conflict with observations in a test set of demonstrations (and thus reduce their likelihood to zero), then we observe the overfitting effect on the constraints themselves (we’ve chosen constraints that do not exist in the true MDP). On the other hand, if we do identify a solution that generalizes perfectly to all new demonstrations, then we have not selected a false constraint that conflicts with a demonstration, and thus not over-fit.
>
> You also mentioned that more real-world experiments would be welcome. While we believe that the presented results do illustrate the usefulness and functionality of the method, we agree that more involved scenarios could better “stress test” the method. However, we have no additional real-world results to report at this time.
>
> Finally, to your point on discussing the limitations of the work, we agree that it is important to be upfront about these points, so we’ve added some discussion on this to the end of sec. 3.3. However, we would like to clarify the point you mention about “constraining the possibility to drive slowly.” In this case, a constraint preventing slow driving would only be learned if the task incentivized slow driving, but none was observed. If the task incentivized fast driving (such as reaching a destination quickly), then it would be unexpected to see slow driving anyway, making it an unlikely constraint candidate (though it would still be a valid candidate).

---

### Official Review · AnonReviewer2 · 2019-10-30
**Official Blind Review #2**

**Rating:** 6

**Review:**

The paper aims to address a new method for inverse reinforcement learning based on maximum likelihood constrained inference. In general, I find the problem very interesting and the motivation of the work is quite reasonable. However, I have two major comments:

(i) Does the constraint semantically similar to the domain of the MDP? In my intuition, one can create a convex hull over the state and action representations to  actually estimate the constraint.

(ii) Suppose, the reward function is unknown, how your method will fare in this case?

**Experience Assessment:**

I have read many papers in this area.

**Review Assessment: Checking Correctness Of Derivations And Theory:**

I did not assess the derivations or theory.

**Review Assessment: Checking Correctness Of Experiments:**

I assessed the sensibility of the experiments.

**Review Assessment: Thoroughness In Paper Reading:**

I read the paper at least twice and used my best judgement in assessing the paper.

---

> ### Author Response · Authors · 2019-11-14
> **Response to Review #2**
>
> Thank you for your review and for pointing out that you find the problem interesting and well-motivated! We address your two comments below, and we believe this discussion will lead to a better paper overall.
>
> “(i) Does the constraint semantically similar to the domain of the MDP? In my intuition, one can create a convex hull over the state and action representations to  actually estimate the constraint.”
> Since we define constraints as sets of state-action pairs which we prohibit, it is possible to interpret the addition of constraints as modifying the domain of the MDP by removing states and/or actions. We believe the idea of considering convex hulls over the state/action space actually aligns well with our set-based notion of constraints. For instance, if you define a feature that indicates whether an agent is inside (or outside) of some convex hull, then the space within this convex hull (or the space outside of it) would be equivalent to the set defining a feature constraint in our formulation. If all demonstrations remain within this convex hull (even though we expect some to exit), then our method can infer that remaining within the convex hull is a constraint.
>
> “(ii) Suppose, the reward function is unknown, how your method will fare in this case?”
> Thank you for mentioning this, as this is a point that we can further clarify. Our algorithm does require some nominal reward function to operate, but this reward function does not need to be perfect. The purpose of the reward function is to inform what behavior we would expect to see, so that we can quantify how demonstrations deviate from these expectations, which allows us to infer the constraints most likely to cause this difference. The more accurate the nominal reward is, the more accurate our expectations will be, leading to better constraint inference.
> 	In the absence of an a priori known reward, we see two possible options to choose / recover a reward so that our method can be used. 1) Use general knowledge / intuition about the task to propose a simple nominal reward (e.g. “I know this is a navigation task, so incentivize short paths” or “I know resources are constrained, so penalize energy usage”). 2) Gather demonstrations from a baseline nominal condition (such as a well-characterized region where all constraints are known beforehand) and use existing IRL techniques to estimate a reward function. Our method can then be used to estimate the unknown constraints in previously unseen conditions sharing that basic reward structure. We followed approach 2) in our human obstacle avoidance example (sec. 4.2), by performing MaxEnt IRL on demonstrations of humans navigating the space without an obstacle. In the novel condition where the obstacle is added, our method detects that the human trajectories deviate from expectations and infers the presence of the previously unknown obstacle.
> 	We’ve added some discussion on this point in a new subsection 3.2.2 that discusses when we might use 1) or 2) to choose a reward if one is not already available.

---

### Official Review · AnonReviewer4 · 2019-11-04
**Official Blind Review #4**

**Rating:** 8

**Review:**

In this work, a novel inverse constraint learning method is proposed, where the goal is to find out the constraints over state-action pairs for given demonstration and MDP **including a reward function** (so different from inverse cost learning). The novelty of this work comes from introducing maximum entropy inverse reinforcement learning (MaxEntIRL) framework to previous works [1, 2], and this work mainly focused on the tabular setting. The objective of this work is to solve the optimization in (8), which tries to find out the constraint that maximizes the probability of trajectories that cannot be generated if that constraint is applied. (Such an objective minimizes the normalization constant in (5) and results in maximization of the demonstration likelihood under the constraint.) To solve this objective, the proposed algorithm first computes the feature occupancy (Algorithm 1), and then use those feature occupancy with greedy iterative constraint inference (Algorithm 2 that motivated by maximum coverage problem) to get constraints. Two experiments in the GridWorld show that the proposed method effectively works.

I think this work is quite fundamental, impactful to be accepted at the conference and is possibly extended to practical scenarios (like explainable and safety RL and imitation learning) in the future. One thing I’d like to point out is to enhance the readability by reordering contents and adding some additional explanations to clarify their arguments. There are a few comments and questions I have:

- The exact definition and usage of nominal environments and rewards are still unclear to me. For example in Figure 2 (b), how did you define and get nominal MDP?
- Since Figure 1 is related to the second experiment, I recommend moving it to the experiment section.
- At 3.2.1., “Because constraints are sets of state-action pairs, imposing a
constraint within an MDP means restricting the set of actions that can be taken from certain states.” -> Need clarification
- At 3.2.1., “For MDPs with deterministic transitions, it is clear that any agent respecting these constraints will not visit an empty state.” -> Why?
- At (8), $\{\}$ to $\emptyset$
- In Figure 3, I was a bit confused about the relationship between the threshold and the false positive rate at first glance. What I understood is that a small threshold leads to lots of iteration for constraint selection, which increases the false positive. I want authors to add some comments on that.

 Reference
[1] Chou, Bereson, Ozay, “Learning Constraints from Demonstrations,” arXiv 2019
[2] Chou, Bereson, Ozay, “Learning Parametric Constraints in High Dimensions from Demonstrations,” CoRL 2019

**Experience Assessment:**

I have published one or two papers in this area.

**Review Assessment: Checking Correctness Of Derivations And Theory:**

I assessed the sensibility of the derivations and theory.

**Review Assessment: Checking Correctness Of Experiments:**

I carefully checked the experiments.

**Review Assessment: Thoroughness In Paper Reading:**

I read the paper thoroughly.

---

> ### Author Response · Authors · 2019-11-14
> **Response to Review #4**
>
> Thank you for your detailed and encouraging review, which provides an excellent summary of our work! We are glad that you find this work impactful, and we appreciate your constructive feedback. We address your comments below, and we believe this discussion will lead to a stronger paper overall.
>
> “- The exact definition and usage of nominal environments and rewards are still unclear to me. For example in Figure 2 (b), how did you define and get nominal MDP?”
> Broadly speaking, the nominal environments and rewards can be thought of as either “generic” or “baseline” conditions.
> We talk about the generic case at the start of section 3.2, where you design a model flexible enough to describe all reasonable car behaviors, then you can use our method to infer the particular constraints applying to a model of car, a type of roadway, a specific driver, or a combination of these specific factors. The nominal model from Figure 2(b) falls into this “generic” category: we assume that we know the basic reward (reaching the goal quickly is incentivized) as well as the structure of the environmental elements (states, features, and actions). However, we assume that we do not know which of these elements our demonstrators might consider constraints, so we learn the particular relationship through their demonstrations.
> The other perspective is to think of the nominal model/reward as a “baseline” condition. If we think of nominal models as coming from a baseline condition, then we would derive them from observations of a system in a known, well-characterized configuration. Our method is then useful for detecting new, unknown constraints that alter the system, such as a road closure or a new obstacle entering the space. For instance, in our human obstacle avoidance example, we first learn a nominal reward from the baseline condition of humans navigating the empty space, and we use the expectations from this baseline (along with the new demonstrations) to detect the presence of a new obstacle (due to deviations from “baseline” behavior). The baseline type of nominal model is useful anytime you want to detect new constraints altering agent behavior.
> We’ve made these points explicit in the updated paper in a new subsection 3.2.2.
>
> “- Since Figure 1 is related to the second experiment, I recommend moving it to the experiment section.”
> In the updated version, we’ve moved the original Figure 1 to the experiment section, and replaced it with another figure showing how constraints affect behavior in sec. 3.2.
>
> “- At 3.2.1., “Because constraints are sets of state-action pairs, imposing a
> constraint within an MDP means restricting the set of actions that can be taken from certain states.” -> Need clarification”
> We’ve rephrased this idea in the updated version to make this clearer, instead stating that:
> If we impose a constraint on an MDP, then none of the state-action pairs in that constraint set may appear in a trajectory of the constrained MDP. To enforce this condition, we must restrict the actions available in each state so that it is not possible to produce one of the constrained state-action pairs.
>
> “- At 3.2.1., “For MDPs with deterministic transitions, it is clear that any agent respecting these constraints will not visit an empty state.” -> Why?”
> The assertion follows from the fact that, with deterministic transitions, agents are able to know exactly the state to which they will transition given their chosen action. If an action deterministically leads to an empty state, then the agent knows that choosing such an action will put them in a position (the next state) where they have no available future actions which respect the constraints. Therefore, any demonstrations from an agent which respects the constraints must also avoid empty states. We’ve reworded this text in the updated version to clarify this point.
>
> “- At (8), {} to $\emptyset$“
> We have followed your suggestion to update our notation for the empty set.
>
> “- In Figure 3, I was a bit confused about the relationship between the threshold and the false positive rate at first glance. What I understood is that a small threshold leads to lots of iteration for constraint selection, which increases the false positive. I want authors to add some comments on that.”
> Yes, your understanding is exactly right on that point. We’ve added some additional comments to the paper that clarify this directly.

---

### Decision · Program_Chairs · 2019-12-19

**Decision:**

Accept (Spotlight)

**Comment:**

The paper introduces a novel way of doing IRL based on learning constraints. The topic of IRL is an important one in RL and the approach introduced is interesting and forms a fundamental contribution that could lead to relevant follow-up work.